# GPS Coordinates for Modelling Correlated Herd Effects in Genomic Prediction Models Applied to Hanwoo Beef Cattle

**DOI:** 10.3390/ani11072050

**Published:** 2021-07-09

**Authors:** Beatriz Castro Dias Cuyabano, Gabriel Rovere, Dajeong Lim, Tae Hun Kim, Hak Kyo Lee, Seung Hwan Lee, Cedric Gondro

**Affiliations:** 1Department of Animal Science, Michigan State University, 474 S Shaw Ln, East Lansing, MI 48824, USA; roverega@msu.edu; 2French National Institute for Agriculture, Food, and Environnement (INRAE), Génétique Animale et Biologie Intégrative (GABI), (Current Institution), 78350 Jouy-en-Josas, France; 3Department of Epidemiology and Biostatistics, Michigan State University, 909 Wilson Rd, East Lansing, MI 48824, USA; 4Center for Quantitative Genetics and Genomics, Aarhus University, Blichers Allé 20, 8830 Tjele, Denmark; 5Animal Genome & Bioinformatics Division, National Institute of Animal Science, RDA, Wanju 55365, Korea; lim.dj@korea.kr (D.L.); thkim63@korea.kr (T.H.K.); 6Department of Animal Biotechnology, Chonbuk National University, Jeonju 54896, Korea; breedlee@jbnu.ac.kr; 7Division of Animal and Dairy Science, Chungnam National University, Daejeon 305764, Korea; slee46@cnu.ac.kr

**Keywords:** genetic evaluation, variance components, carcass traits, geographical location

## Abstract

**Simple Summary:**

It is widely known that the environment influences phenotypic expression and that its effects must be accounted for in genetic evaluation programs. The most used method to account for environmental effects is to add herd and the contemporary group to the model. Although generally informative, the herd effect treats different farms as independent units. However, if two farms are located physically close to each other, they potentially share correlated environmental factors. We introduce a method to model herd effects using physical distances between farms based on GPS coordinates as a proxy for the correlation matrix of these effects, aiming to account for similarities and differences between farms due to environmental factors. A population of beef cattle was used to evaluate the impact on the variance components and on the genomic prediction, of modelling herd effects as correlated, in comparison to assuming the farms as completely independent units. The main result was an increase in the reliabilities of the predicted genomic breeding values compared to reliabilities obtained with traditional models, a finding of practical relevance for genetic evaluation programs.

**Abstract:**

It is widely known that the environment influences phenotypic expression and that its effects must be accounted for in genetic evaluation programs. The most used method to account for environmental effects is to add herd and contemporary group to the model. Although generally informative, the herd effect treats different farms as independent units. However, if two farms are located physically close to each other, they potentially share correlated environmental factors. We introduce a method to model herd effects that uses the physical distances between farms based on the Global Positioning System (GPS) coordinates as a proxy for the correlation matrix of these effects that aims to account for similarities and differences between farms due to environmental factors. A population of Hanwoo Korean cattle was used to evaluate the impact of modelling herd effects as correlated, in comparison to assuming the farms as completely independent units, on the variance components and genomic prediction. The main result was an increase in the reliabilities of the predicted genomic breeding values compared to reliabilities obtained with traditional models (across four traits evaluated, reliabilities of prediction presented increases that ranged from 0.05 ± 0.01 to 0.33 ± 0.03), suggesting that these models may overestimate heritabilities. Although little to no significant gain was obtained in phenotypic prediction, the increased reliability of the predicted genomic breeding values is of practical relevance for genetic evaluation programs.

## 1. Introduction

The concept of estimated breeding values and the development of best linear unbiased prediction (BLUP) methodology [1,2] ushered in the modern approach to genetic evaluation in animal breeding through the integration of pedigree information to calculate genetic relationships between animals [3]. This allowed the inclusion of performance measures of relatives to estimate the breeding value of an individual—the animal model. However, it was only in the late 1980s that computers became powerful enough to enable the use of animal models for genetic evaluation in practice. From there on, BLUP became widely adopted as a tool to estimate breeding values of animals for livestock production and grew into the cornerstone of successful genetic improvement programs of livestock populations [4].

The accuracy of genetic evaluations using BLUP relies on the pedigree relationships between individuals [5,6] and on the connections between contemporary groups [7]. With the development of genomic technologies and models that incorporate dense marker information [8], these genetic evaluations moved from relying solely on the pedigree relationships between individuals, to relying on the realized genetic relationships between individuals [9] estimated from their genetic markers. These genomic relationship matrices (GRM) can then be used instead of the traditional pedigree relationship matrix in a BLUP extension termed genomic best linear unbiased prediction—GBLUP [9]. In practice, industry largely uses both pedigree and genomic information simultaneously in a combined model known as single step GBLUP [10].

Genetic evaluation models largely assume that economically important traits are quantitative, that is, that the traits are influenced by a large number of genes. Many of these models also try to account for the various environmental effects to which animals are exposed. The common method to consider the effects of the different environments is to include herd and the contemporary group in the model [6,11]. Although there is a case to fit the effects of the contemporary groups (CG) as fixed [12], there are strong arguments to fit them as random effects in models for genomic prediction [13]. Henderson did conceptualize CG effects as random, however opted to treat CG as fixed under the argument that it eliminated bias from the sire model [12]. When sire models were widely applied, the choice of fixed CG effects was sensible. Nowadays, most genetic evaluation systems no longer consider the sire model, preferring the animal model instead. Nonetheless, CG effects often continue to be treated as fixed, even though in the animal model CG could be treated as random without compromising BLUP’s unbiasedness property [13]. Moreover, when CG are small, or when animals have relatively low genetic connections, fitting the CG as random can improve the accuracy of prediction, compared to fitting CG as fixed [14,15,16,17,18]. Traditionally, the herd effect is modeled by assuming different farms as completely independent units. However, if two farms are physically close to each other, it is not unrealistic to expect that some of the climate, geographical factors, management strategies, and even social aspects of production would be more similar between these two farms than between two farms farther apart from each other. Treating farms as non-independent units in genomic prediction is not yet a usual practice. In a study aiming to estimate genotype by environment (GxE) interactions, management strategies, information about the CG, and climate data have been used to create correlated environmental information to be included in the model for genomic prediction [19]. A more recent study that accounted for the geographical location of farms indicated that adding a correlation structure between herd effects benefits the genetic evaluation in smallholder breeding programs [20].

Spatial modelling is widely used in geostatistics studies [21,22,23]; however, the use of spatial models is still limited in the context of animal breeding. Most spatial models are designed with the objective of making inferences for regions where no observations have been previously collected. For such scenarios, a Gaussian random field [24] is a popular choice, with the Matérn covariance function [25] being the most commonly used structure to model spatial covariance.

Here we introduce a method for genomic prediction that models the herd effects through a covariance matrix derived directly from the Euclidian distances between farms based on their Global Positioning System (GPS) coordinates, which serves as a proxy for the underlying unknown environmental correlation matrix between herds. Using the physical distances between individual farms, we defined a herd covariance matrix that enabled the model to account for their similarities or differences due to climate and other environmental factors, when these factors are not explicitly measured. The objective of this study was to evaluate how modelling the herd effects as correlated impacts the variance components and the genomic prediction, in comparison to assuming the farms as completely independent units.

## 2. Material and Methods

The Animal Care and Use Committee of the National Institute of Animal Science (NIAS), Rural Development Administration (RDA), South Korea, approved the experimental procedures (data used is part of project with approval no. 2018-293), and appropriate animal health and welfare guidelines were followed.

### 2.1. Genotypes, Phenotypes, and GPS Data

We used industry production trait data from a population of Korean Hanwoo cattle to compare our proposed model with models that did not include the GPS information. Genomic and phenotypic data were available for a total of 4168 commercial animals sampled throughout South Korea from a population of the Hanwoo beef cattle breed. Individuals were genotyped using the 50 k Illumina (San Diego, CA, USA) Bovine SNP50V2 BeadChip array, and after quality control filtering (SNPs with minor allele frequency <0.01 and individuals with >2% missing genotypes were excluded), 43,749 SNP genotypes were included in the analysis. These were commercial and genetically unrelated animals (no pedigree was available to these animals, and relationships in the genomic relationship matrix were low), spread across 124 finishing farms in South Korea. Table 1 summarizes the number of animals sampled across the farms. All 124 farms had their GPS coordinates described by their latitude and longitude, and Figure 1 shows their dispersion in South Korea along with a summary about the locations’ altitude, average temperature, and average cumulative precipitation (historical means on measurements between the years 2000 and 2019).

All animals were born between 2014 and 2015 and slaughtered in 2017, at ages that ranged from 25 to 35 months. The data set comprised only male animals that were classified as bulls or steers. Beef production traits were recorded, and we performed our analyses on four of these traits: backfat thickness (BFT), eye muscle area (EMA), carcass weight (CWT), and marbling score (MS). BFT was measured in millimeters at the cross-sectional slice between the thoracic and the 1st lumber vertebrae perpendicular to the vertebral column, EMA was also measured from the same region in squared centimeters using a dot-grid; carcass weight was measured in kilograms at the end of refrigeration for 24 h after slaughter; marbling scores were visually classified according to standard grading guidelines into nine ordinal levels [28,29]. Table 2 presents a summary of the traits and number of animals per sex and age group.

### 2.2. Prediction Models

Five prediction models were compared, with their equations fully described in Table 3. Each model consisted of the fixed effects (denoted along with their design matrix as Xb), and of a varying combination of the breeding values (g) and the herd effects (η), in our analyses represented by the farms, either assuming the farms as completely independent units ($\eta_{FARM}$) or correlated by their geographical distances ($\eta_{GPS}$). We assumed herd effects as random in all models that accounted for these effects. Models GRM, FARM, and GPS fitted each of these three random components alone, and models GRM + FARM and GRM + GPS fitted the breeding values with the herd effects simultaneously, with the two components (breeding values and herd effects) fitted as independent from one another.

A preliminary study indicated that age (fitted as a continuous covariate), sex (bull/steer), and size of the contemporary group of the herd (fitted as a continuous covariate, with size being the number of animals from each herd) were statistically significant enough to be included as fixed effects in all the models. Year, month, and season of birth were not found to be significant effects in our models. Our data consisted of animals slaughtered in the same year (2017), therefore year of slaughter was not included in the model. Although animals were slaughtered from March 2017 to November 2017, the vast majority (~65%) were slaughtered in the summer months (July–September), and neither season nor month were found to be significant effects.

Breeding values were assumed to be distributed $g∼N\left(0,G\sigma_{g}^{2}\right)$, where G denoted the genomic relationship matrix (GRM) as per [9]. Herd effects were considered random in all models that accounted for these effects, with Z being the design matrix indicating the herd to which the phenotypic observations y belonged to. Models FARM and GRM + FARM assumed the herd effects as independent with distribution $\eta_{FARM}∼N\left(0,I_{F}\sigma_{\eta }^{2}\right)$, where $I_{F}$ is an identity matrix of order F, the number of herds. Models GPS and GRM + GPS assumed the herd effects as correlated with distribution $\eta_{GPS}∼N\left(0,E\sigma_{\eta }^{2}\right)$, in which E is the herd covariance matrix, obtained with standardized distances between the farms on which each individual was located derived from their GPS coordinates. The following steps were used to obtain the E matrix: First, a matrix D was calculated with the raw pairwise distances between farms. This was performed using the function distm from the R [30] package geosphere [31], setting the argument fun = distGeo. The second step was to rescale the distances and ensure that all values were between zero and one by dividing them by the largest distance observed, $D^{*}=D/\max\left\{D_{ij}:i,j=1,\ldots ,F\right\}$. Finally, each element of E was defined as $E_{ij}=1-D_{ij}^{*}$, with values $0<E_{ij}<1$, for every i,j=1,…,F. The closer the farms, the closer to one $E_{ij}$ will be. Likewise, the farther the farms, the closer to zero $E_{ij}$ will be. Random residuals (ε) were assumed to be distributed $\varepsilon ∼N\left(0,I_{n}\sigma_{\varepsilon }^{2}\right)$, where $I_{n}$ is an identity matrix of order n, the number of individuals. In addition, $\sigma_{g}^{2}$, $\sigma_{\eta }^{2}$, and $\sigma_{\varepsilon }^{2}$ denoted the total genomic, herd and residual variances, respectively.

In our study, we did not aim to infer herd effects to herds that were not in the reference population. Therefore, kernels commonly used in spatial analysis and geostatistics to infer unobserved data were not the target kernels to correlate herd effects in our study. That is because we accounted all herds as observed in the reference population, and therefore the herd effects estimated using, for example, the Matérn covariance are mathematically equivalent to the herd effects estimated using the herd covariance matrix based on the standardized distances between the farms (Appendix A). Nonetheless, we ran a set of analyses considering the Matérn covariance as the kernel to correlate herd effects, to confirm empirically the theoretical equivalence of the kernels, and to verify that results would match those obtained using the herd covariance matrix based on the standardized distances between the farms (Appendix A; results presented in Appendix A).

### 2.3. Methods for Variance Components Estimation and Model Assessment

Variance components ($\sigma_{g}^{2}$, $\sigma_{\eta }^{2}$, and $\sigma_{\varepsilon }^{2}$) were estimated on a training group using the restricted maximum likelihood (REML) [32], and genomic prediction was performed using GBLUP [9]. All analyses were conducted using the R [30] programming language, with functions implemented by the authors in the GenEval package, available online at https://github.com/bcuyabano/GenEval.

The (narrow sense) heritability was defined as $h^{2}=\sigma_{g}^{2}/\sigma_{y}^{2}$, where $\sigma_{y}^{2}$ represents the phenotypic variance ($\sigma_{y}^{2}=\sigma_{g}^{2}+\sigma_{\varepsilon }^{2}$ for model GRM, $\sigma_{y}^{2}=\sigma_{\eta }^{2}+\sigma_{\varepsilon }^{2}$ for models FARM and GPS, and $\sigma_{y}^{2}=\sigma_{g}^{2}+\sigma_{\eta }^{2}+\sigma_{\varepsilon }^{2}$ for models GRM + FARM and GRM + GPS). The proportion of phenotypic variance due to the herd was defined as $e^{2}=\sigma_{\eta }^{2}/\sigma_{y}^{2}$. We will from now on refer to $e^{2}$ as environmentability.

The performance of the different models for genomic prediction was assessed considering the reliability of the predicted genomic breeding values (PGBV) (${\tilde{g}}_{test}$) and of the predicted herd effects (${\tilde{\eta }}_{test}$) on a test group (here, ${\tilde{\eta }}_{test}$ refers to either ${\tilde{\eta }}_{FARM,test}$ or ${\tilde{\eta }}_{GPS,test}$, depending on whether the herd model matrix $I_{F}$ or E was used). The reliabilities were calculated as $R_{PGBV}^{2}=r_{PGBV}^{2}/h^{2}$ and $R_{\eta }^{2}=r_{\eta }^{2}/e^{2}$, a proxy for the average individual reliabilities of each breeding value, such that $r_{PGBV}=cor\left({\tilde{g}}_{test},y_{test}-X_{test}\hat{b}\right)$ and $r_{\eta }=cor\left(Z{\tilde{\eta }}_{test},y_{test}-X_{test}\hat{b}\right)$ are the accuracies of prediction of the genomic breeding values and herd effects, respectively. The values of $R_{PGBV}^{2}$ and $R_{\eta }^{2}$ are constrained to the (0,1) interval, and desired to be as high as possible. We also compared the accuracy of the predicted phenotypes, calculated as $r_{y}=cor\left({\tilde{y}}_{test},y_{test}\right)$. The predicted phenotypes were ${\tilde{y}}_{test}=X_{test}\hat{b}+{\tilde{g}}_{test}$ for the GRM model, ${\tilde{y}}_{test}=X_{test}\hat{b}+Z{\tilde{\eta }}_{test}$ for the FARM and the GPS models, and ${\tilde{y}}_{test}=X_{test}\hat{b}+{\tilde{g}}_{test}+Z{\tilde{\eta }}_{test}$ for the GRM + FARM and GRM + GPS models. Each model was replicated 100 times, with each replicate consisting of reassigning 80% of the individuals to the training group and the other 20% of the individuals to the test group. Tukey’s multiple comparison test [33] was used to compare the results obtained with the different models within each trait, at a significance level of 0.05. To perform the contrast between models within each trait, the prediction accuracies and reliabilities (after taking their square root) were normalized using Fisher’s z-transformation [34]. Since the heritabilities were obtained using REML, they are therefore assumed to be normally distributed, thus no transformation was done to their values. To compare the PGBVs obtained with models GRM + FARM and GRM + GPS to the PGBVs obtained with model GRM, Pearson and Spearman correlations were calculated, as well as the mean squared differences between the PGBVs, defined as $MSD_{model\;1,model\;2}=1/n_{test}\overset{n_{test}}{\underset{i=1}{\sum }}{\left({\tilde{g}}_{i,model\;1}-{\tilde{g}}_{i,model\;2}\right)}^{2}$.

To rule out any potential confounding effects between G and $W_{\eta_{FARM}}=ZZ\prime$, and between G and $W_{\eta_{GPS}}=ZEZ\prime$, we compared their variance structures using the kappa statistics ($\kappa_{\eta :GRM}$) plotted against the eigen-values ($\lambda_{\eta }$) of $W_{\eta_{FARM}}$ and $W_{\eta_{GPS}}$, as proposed by [5]. If $\kappa_{\eta :GRM}$ and $\lambda_{\eta }$ are very aligned, there is confounding between the matrices. If $\kappa_{\eta :GRM}\approx 1$, there is no confounding between the matrices [35]. Each element of $\kappa_{\eta :GRM}$ was computed as $\kappa_{\eta :GRM,i}=\overset{n}{\underset{j=1}{\sum }}{\left(U_{\eta ,i}^{\prime }U_{GRM,j}\right)}^{2}\lambda_{GRM,j}$ for every i=1,…,n (n being the number of observations), where $U_{\eta ,i}$ and $U_{GRM,j}$ are the $i^{th}$ and $j^{th}$ eigen-vectors of $W_{\eta }$ (FARM or GPS) and G, respectively, and $\lambda_{GRM,j}$ is the $j^{th}$ eigen-value of G.

## 3. Results

Figure 2 presents the principal components plot of the genomic relationship matrix G, and this plot shows a grouping of individuals into two main genomic groups; however, these groups are not associated with any particular herd.

Table 4 presents the values obtained for heritability, environmentability, prediction accuracies, and reliabilities for each of the traits and different models. The statistical differences between models within a trait were tested using Tukey’s multiple comparison test, at a significance level of 0.05. Results will be presented, compared, and discussed based on the groups obtained with the multiple comparison test. Therefore, when we state that values are equal, lower, or higher, we mean statistically equal, lower, or higher at a significance level of 0.05.

Heritability estimates obtained for EMA, CWT, and MS obtained with the GRM + GPS model were lower than those obtained with the GRM and GRM + FARM models. Heritability estimates obtained for BFT were equal in all three models that included breeding values. The environmentability estimates obtained with the GPS and GRM + GPS models were higher than those obtained with the FARM and GRM + FARM models for all traits. Environmentability estimates obtained with the FARM and GRM + FARM models were equal for BFT and EMA, while environmentabilities estimated with the FARM model were higher than those estimated with the GRM + FARM model for CWT and MS. Except for CWT, which had a higher environmentability estimate with the GPS model than with the GRM + GPS model, environmentabilities estimated with the GPS and GRM + GPS models were equal.

Prediction accuracies of the PGBV obtained with all the models were equal for all traits. Table 5 presents the Pearson and Spearman correlations and the mean squared differences used to compare the PGBVs obtained with models GRM + FARM and GRM + GPS to the PGBVs obtained with model GRM. Although both the Pearson and the Spearman correlations were consistently higher for model GRM + GPS, their values were between 0.97 and 1 for both the GRM + FARM and GRM + GPS models, indicating that including the herd effects to the genomic prediction did not produce significant changes in the ranking of the PGBVs, when compared to the GRM model that did not account for the herd effects. Comparing the PGBVs by the mean squared differences, we observed that the GRM + GPS model obtained PGBVs more similar to those obtained with the GRM model than did the GRM + FARM model. This indicates that in our data, the inclusion of correlated herd effects interfered less with the PGBVs than did the inclusion of independent herd effects.

Prediction accuracies of the herd effects obtained with the GPS model were higher than those obtained with the FARM model for all traits. Except for CWT that had equal herd effect prediction accuracy with the GRM + GPS and GRM + FARM models, prediction accuracies of the herd effects obtained with the GRM + GPS model were higher than those obtained with the GRM + FARM model. Prediction accuracies of the herd effects obtained with the FARM and GRM + FARM models were equal for BFT and CWT, while prediction accuracies of herd effects obtained with the FARM model were lower than those obtained with the GRM + FARM model for EMA and MS. Prediction accuracies of the herd effects obtained with the GPS and GRM + GPS models were equal for all traits.

Reliabilities of PGBV and environmental effects are shown in Figure 3. All PGBV reliabilities from the GRM + GPS model were higher than those from the GRM and GRM + FARM models, and the difference is particularly large for EMA, CWT, and MS. The PGBV reliabilities between the GRM and the GRM + FARM models are equal except for MS. Reliabilities of the predicted herd effects were higher in the FARM and GRM + FARM models than in the GPS and GRM + GPS models, except for MS that was the opposite.

Figure 4 shows the accuracy of phenotypic prediction. The inclusion of a herd effect component in the genomic model provided a significant phenotypic prediction gain for BFT, EMA, and MS. For BFT, this gain was observed only with the GRM + GPS model. Additionally, BFT was the only trait in which we observed a significant phenotypic prediction gain by using the GRM + GPS model instead of the GRM or GRM + FARM models; however, it was a small gain: the mean observed $r_{y}$ with the GRM, GRM + FARM, and GRM + GPS models were respectively 0.372, 0.363, and 0.383. From the boxplots of the accuracy of phenotypic prediction we could observe that, for BFT and EMA, the prediction accuracies obtained over the 100 replicates with the model GRM + GPS varied less than the prediction accuracies obtained with the model GRM + FARM.

Figure 5 presents the comparison of the variance structures between G and $W_{\eta_{FARM}}=ZZ\prime$, and between G and $W_{\eta_{GPS}}=ZEZ\prime$, using the kappa statistics [35]. We observed that $\kappa_{\eta :GRM}$ and $\lambda_{\eta }$ are not aligned at all, and that $\kappa_{\eta :GRM}\approx 1$, indicating that there is no confounding between the two relationship matrices.

## 4. Discussion

This work evaluated how modelling the herd effects as correlated impacts the variance components and the genomic prediction, in comparison to assuming the herds as completely independent units. To estimate and predict the correlated herd effects, we presented a method that used the physical distances (based on GPS coordinates) to create a herd covariance matrix, which was used as a proxy to model the unknown underlying correlation matrix of these effects. Conceptually, this herd covariance matrix should enable the model to partially account for similarities or differences between herds due to climate and other geographical factors that were not explicitly measured and included in the model. We tested this assumption on four production traits in a commercial population of Hanwoo beef cattle sampled from across South Korea by modelling herd effects using GPS coordinates. The main result was an increase of the reliabilities of PGBV for all traits in comparison to the reliabilities obtained with a model that assumed the herd effects to be independent or with a model that did not account for the herd effects at all.

The source of this increase in the reliabilities of PGBV is of particular interest. It is basically because the GRM + GPS model reallocated to the herd effect a proportion of the phenotypic variance that the other models assigned to the breeding values. In all four traits, the heritability estimates of the GRM + GPS model were significantly lower than the estimates of the GRM and GRM + FARM models. The decrease in the heritability estimates led to an increase in the PGBV reliabilities since $R_{PGBV}^{2}=r_{PGBV}^{2}/h^{2}$.

At this point, our results suggest that the heritabilities of the traits estimated with the GRM + GPS model are more accurate than the higher values estimated by both the GRM and GRM + FARM models. If the GRM + GPS model was underestimating the heritabilities in comparison to GRM and GRM + FARM models, we would expect changes in the PGBV variances or a reduction in the prediction accuracies. However, across all four traits, the PGBV variances and the prediction accuracies were statistically equal for the GRM, GRM + FARM, and GRM + GPS models. The reduction in heritability estimates with the GRM + GPS model did not affect the PGBV accuracies, which suggests that the GRM and GRM + FARM models probably overestimated the variance of the breeding values. If the PGBV reliabilities are in fact higher than what traditional approaches indicate, there are relevant real-world implications for breeding programs, since accurate estimates are of practical importance and are broadly used to design breeding programs, predict response to selection, and determine selection strategies [36,37,38].

The covariance matrix built from GPS coordinates to model herd effects was expected to account for some of the similarities or differences between herds due to climate and other environmental factors. This was confirmed by the much higher environmentability estimates from the GPS and GRM + GPS models compared to the environmentabilities estimated with the FARM and GRM + FARM models. Even though the GRM + GPS reallocated genetic variance into environmental variance, the total variance explained was much higher than with the other models (Table 4), which would intuitively suggest a better predictive value as well; however, as predictors, the GPS information was not sufficiently informative to improve phenotypic prediction; little to no significant gain in phenotypic prediction was achieved with the GRM + GPS model. In part, this could be due to the data used in this study being relatively small and to South Korea being also a small country with a relatively uniform climate and production environment. It is possible that prediction accuracies would increase with a larger dataset from a country with more diverse conditions. It is, however, more likely that to substantially improve phenotypic prediction, direct and detailed environmental/management data modelling will be necessary.

Although the results from our study suggest that herd effects should be modelled as correlated, we understand that the herd covariance matrix based on GPS coordinates is just an approximation that tries to simplistically capture a very wide range of environmental and farm conditions. While modelling herds as correlated effects clearly accounted for significant amounts of phenotypic variation, and it did not negatively impact genomic prediction accuracies, it was still not descriptive enough to increase prediction accuracies in this study. Different methods to correlate herd effects based on GPS coordinates should be further explored in follow-up studies. Moreover, recent studies using different approaches for phenotypic prediction have incorporated climate information measured in more diverse environments than what we observed in South Korea, and have reported positive gains in prediction accuracy in livestock [39] and in crops [40,41]. We believe that the inclusion of environmental parameters to quantify their influence on phenotypic variance and prediction accuracies and to understand how they correlate with the GPS herd covariance matrix can further improve models for genomic and phenotypic prediction.

Finally, although the models with a herd effect that we explored in this study provided little improvement in phenotypic prediction, the environmentability estimates and the reliabilities of the PGBV achieved with the models that used the GPS coordinates are relevant for genetic evaluations. Modelling herd effects as correlated instead of independent indicated that results from traditional models might overinflate the heritability estimates. These overestimates can lead to wrong reliabilities of the PGBV and, consequently, to misleading values of expected genetic gains. In this study, we modelled the correlations between herds through their physical distances using GPS coordinates, but other methods and direct environmental factors such as differences in altitude, temperature, precipitation, or even management or nutrition could be considered instead.

## 5. Conclusions

Environmental effects have a large influence on phenotypic expression and must be accounted for in genetic evaluations. This is usually achieved through the inclusion of a herd effect term in the model. Accurately modelling these herd effects is fundamental to adequately assign phenotypic variance to breeding values and random herd effects, in order to maximize prediction accuracy and reliability of the PGBV. Our results showed that modelling herd effects by considering the herds as being correlated with each other was more successful in capturing environmental variance (environmentability) and the reliability of the PGBV was greater than when herds were considered as completely independent units. The increased reliabilities of the PGBV are particularly relevant for decision making in breeding programs.

## Figures and Tables

**Figure 1 animals-11-02050-f001:**
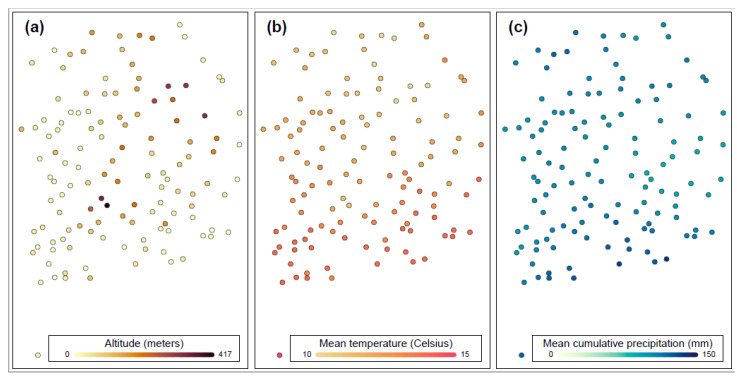
Dispersion of the Hanwoo farms in South Korea with a summary about the locations’ (**a**) altitude (1–147 m), (**b**) average temperature (10.0–15.3 °C), and (**c**) average cumulative precipitation (83.2–138.3 mm). Average temperature and precipitation are historical means on measurements between the years 2000 and 2019 [26,27].

**Figure 2 animals-11-02050-f002:**
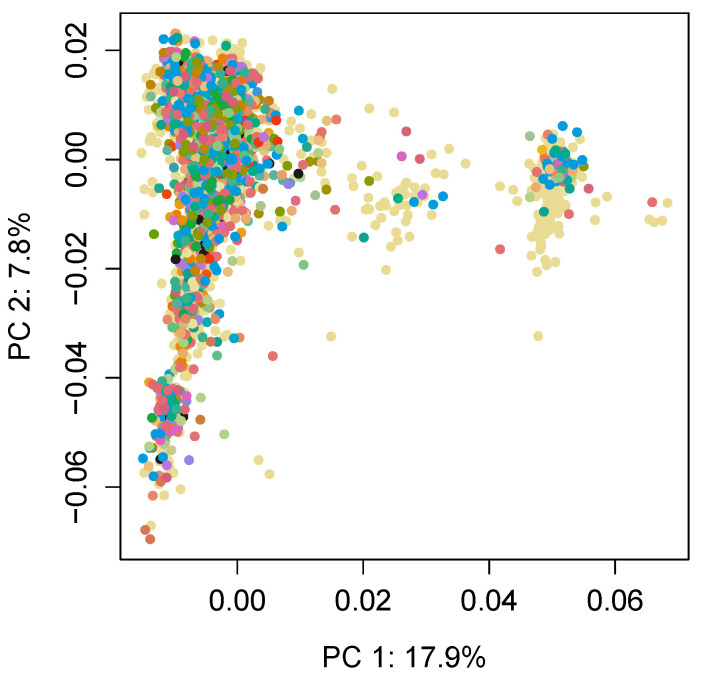
Principal components analysis of the population based on the genomic relationship matrix (G). The different colors indicate the 124 farms on which each individual was located.

**Figure 3 animals-11-02050-f003:**
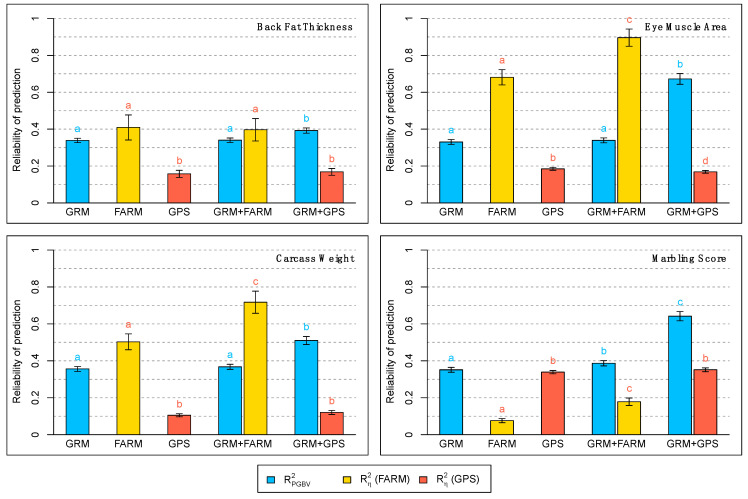
Reliability of predicted genomic breeding values ($R_{PGBV}^{2}=r_{PGBV}^{2}/h^{2}$), and reliability of predicted herd effects ($R_{\eta }^{2}=r_{\eta }^{2}/e^{2}$), for all four traits and all the five models considered. The heights of the barplots in this figure represent the mean over 100 replicates of each model, with their confidence intervals indicated. Letters a, b, c, d indicate the groups obtained with Tukey’s multiple comparison test at a significance level of 0.05, comparing the values of $R_{PGBV}^{2}$ (blue letters) and of $R_{\eta }^{2}$ (red letters) obtained with the different models, within each trait.

**Figure 4 animals-11-02050-f004:**
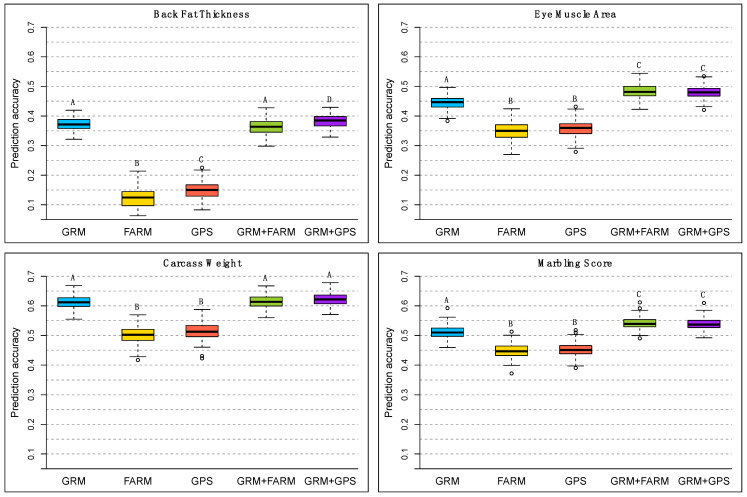
Accuracy of phenotypic prediction ($r_{y}=cor\left({\tilde{y}}_{test},y_{test}\right)$), for all four traits and all the five models considered. Boxplots based on 100 cross validation replicates of each model. The predicted phenotypes were ${\tilde{y}}_{test}=X_{test}\hat{b}+{\tilde{g}}_{test}$ for the GRM model, ${\tilde{y}}_{test}=X_{test}\hat{b}+{\tilde{\eta }}_{test}$ for the FARM and the GPS models, and ${\tilde{y}}_{test}=X_{test}\hat{b}+{\tilde{g}}_{test}+{\tilde{\eta }}_{test}$ for the GRM + FARM and GRM + GPS models. Letters A, B, C, D indicate the groups obtained with Tukey’s multiple comparison test at a significance level of 0.05, comparing the values of $r_{y}$ obtained with the different models, within each trait.

**Figure 5 animals-11-02050-f005:**
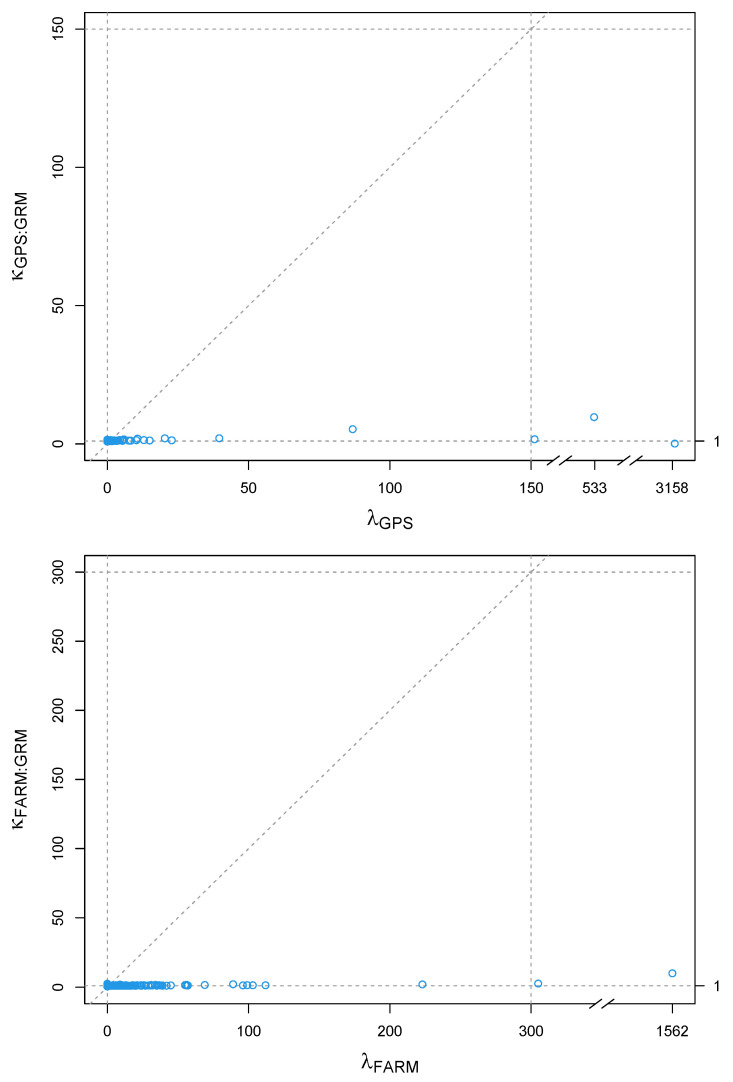
Scatterplots of eigen-values of the herd covariance matrix ($\lambda_{\eta }$) by the kappa statistics ($\kappa_{\eta :GRM}$), in order to compare the variance structures of G and $W_{\eta_{FARM}}=ZZ\prime$, and of G and $W_{\eta_{GPS}}=ZEZ\prime$.

**Table 1 animals-11-02050-t001:** Summary of the number of Hanwoo cattle sampled per farm.

Sample Size	Number of Farms
<5	41
6–10	34
11–20	14
21–50	22
51–100	8
101–350	4
1562	1

**Table 2 animals-11-02050-t002:** Summary of traits (mean ± standard deviation) and number of animals (N), detailed by age, sex, and on all observations (in bold). Traits evaluated were recorded for all animals available for this study.

Trait	Sex	Age (Months)											
		25	26	27	28	29	30	31	32	33	34	35	All
Backfat Thickness (mm)	Bull	12.9 ±4.4	12.8 ±2.7	13.9 ±5.0	12.1 ±3.7	15.1 ±6.2	14.0 ±5.1	14.2 ±4.2	13.1 ±4.8	12.6 ±3.8	15.3 ±5.7	14.2 ±5.7	13.9 ±5.0
	Steer	12.7 ±5.5	13.2 ±5.4	13.3 ±4.8	14.0 ±5.0	14.0 ±4.8	14.6 ±5.1	14.4 ±5.1	15.0 ±5.0	14.5 ±5.1	14.1 ±6.0	14.9 ±6.2	14.4 ±5.1
	All	12.8 ±5.0	13.1 ±4.9	13.4 ±4.8	13.7 ±4.9	14.1 ±4.8	14.6 ±5.1	14.4 ±5.0	14.9 ±5.0	14.3 ±5.0	14.5 ±5.9	14.7 ±6.0	**14.3** **±5.1**
Eye Muscle Area (cm^2^)	Bull	81.7 ±16.7	84.1 ±12.1	89.7 ±15.4	85.5 ±12.4	89.2 ±8.5	84.1 ±10.8	86.5 ±8.8	86.7 ±13.7	87.6 ±12.0	90.7 ±10.9	90.5 ±14.6	87.8 ±12.2
	Steer	87.6 ±14.1	92.7 ±16.7	95.2 ±11.7	97.8 ±12.6	96.4 ±11.6	98.3 ±11.6	98.7 ±11.2	97.5 ±11.4	96.7 ±11.2	96.8 ±11.6	99.0 ±12.8	97.6 ±11.7
	All	85.5 ±15.0	90.9 ±16.1	93.8 ±12.9	96.1 ±13.3	96.0 ±11.5	97.9 ±11.8	98.2 ±11.4	96.9 ±11.9	95.6 ±11.7	94.7 ±11.7	96.0 ±14.0	**96.7** **±12.1**
Carcass Weight (kg)	Bull	319.9 ±76.8	339.9 ±44.4	344.9 ±52.1	344.1 ±44.4	362.0 ±44.5	338.7 ±47.1	350.8 ±25.6	355.5 ±47.9	360.2 ±44.6	379.9 ±48.2	367.6 ±53.0	356.2 ±48.3
	Steer	406.9 ±50.7	418.4 ±57.4	430.2 ±48.7	444.5 ±47.5	445.4 ±46.9	450.2 ±46.1	455.4 ±45.9	456.7 ±49.8	452.9 ±48.5	448.4 ±58.1	465.2 ±59.4	450.1 ±48.2
	All	375.9 ±73.4	402.4 ±63.2	408.2 ±62.0	430.9 ±58.2	440.9 ±50.4	447.1 ±49.7	450.8 ±50.0	450.3 ±55.4	441.2 ±57.0	425.4 ±63.7	430.5 ±73.8	**442.0** **±54.9**
Marbling Score (9 levels)	Bull	4.6 ±2.1	4.5 ±2.2	5.0 ±2.2	4.7 ±1.9	5.3 ±1.6	5.2 ±2.2	5.1 ±2.2	4.1 ±1.9	4.4 ±1.9	5.5 ±1.8	4.9 ±2.0	4.9 ±2.0
	Steer	5.1 ±1.8	6.4 ±2.1	6.4 ±1.8	6.8 ±1.6	6.5 ±1.7	6.7 ±1.6	6.6 ±1.6	6.5 ±1.6	6.4 ±1.6	6.6 ±1.7	6.5 ±1.7	6.6 ±1.7
	All	4.9 ±1.9	6.0 ±2.2	6.1 ±2.0	6.5 ±1.8	6.4 ±1.8	6.7 ±1.6	6.6 ±1.7	6.4 ±1.8	6.1 ±1.8	6.2 ±1.8	5.9 ±2.0	**6.4** **±1.8**
N	Bull	10	10	34	41	44	27	39	30	31	54	38	358
	Steer	18	39	98	261	775	925	855	448	215	107	69	3810
	All	28	49	132	302	819	952	894	478	246	161	107	**4168**

**Table 3 animals-11-02050-t003:** Models used for prediction.

**Model**	**Equation**
GRM	y=Xb+g+ε
FARM	$y=Xb+Z\eta_{FARM}+\varepsilon$
GPS	$y=Xb+Z\eta_{GPS}+\varepsilon$
GRM + FARM	$y=Xb+g+Z\eta_{FARM\;}+\varepsilon$
GRM + GPS	$y=Xb+g+Z\eta_{GPS\;}+\varepsilon$

**Table 4 animals-11-02050-t004:** Heritability and environmentability estimates (${\hat{h}}^{2}=\sigma_{g}^{2}/\sigma_{y}^{2}$ and ${\hat{e}}^{2}=\sigma_{\eta }^{2}/\sigma_{y}^{2}$), prediction accuracy of genomic breeding values and herd effects ($r_{PGBV}=cor\left({\tilde{g}}_{test},y_{test}\right)$ and $r_{\eta }=cor\left({\tilde{\eta }}_{test},y_{test}\right)$), reliability of predicted genomic breeding values and herd effects ($R_{PGBV}^{2}=r_{PGBV}^{2}/{\hat{h}}^{2}$ and $R_{\eta }^{2}=r_{\eta }^{2}/{\hat{e}}^{2}$), for all four traits and all the five models evaluated. Values in this table represent the mean over 100 cross-validation replicates of each model.

Trait	Model	${\hat{h}}^{2}$	${\hat{e}}^{2}$	$r_{PGBV}$	$r_{\eta }$	$R_{PGBV}^{2}$	$R_{\eta }^{2}$
Backfat Thickness	GRM	0.35 ^a,†^	-	0.34 ^a^	-	0.34 ^a^	-
	FARM	-	0.03 ^a,†^	-	0.10 ^a^	-	0.32 ^a^
	GPS	-	0.14 ^b,†^	-	0.14 ^b^	-	0.13 ^b^
	GRM + FARM	0.35 ^a,†^	0.03 ^a,†^	0.34 ^a^	0.10 ^a^	0.34 ^a^	0.32 ^a^
	GRM + GPS	0.30 ^a,†^	0.15 ^b,†^	0.34 ^a^	0.15 ^b^	0.39 ^b^	0.15 ^b^
Eye Muscle Area	GRM	0.35 ^a,†^	-	0.34 ^a^	-	0.33 ^a^	-
	FARM	-	0.09 ^a,†^	-	0.24 ^a^	-	0.66 ^a^
	GPS	-	0.53 ^b,†^	-	0.31 ^b^	-	0.18 ^b^
	GRM + FARM	0.34 ^a,†^	0.09 ^a,†^	0.34 ^a^	0.28 ^c^	0.34 ^a^	0.87 ^c^
	GRM + GPS	0.17 ^b,†^	0.53 ^b,†^	0.34 ^a^	0.30 ^b^	0.66 ^b^	0.17 ^d^
Carcass Weight	GRM	0.41 ^a,†^	-	0.38 ^a^	-	0.35 ^a^	-
	FARM	-	0.06 ^a,†^	-	0.17 ^a^	-	0.47 ^a^
	GPS	-	0.40 ^b,†^	-	0.20 ^b^	-	0.10 ^b^
	GRM + FARM	0.39 ^a,†^	0.05 ^c,†^	0.38 ^a^	0.17 ^a^	0.36 ^a^	0.67 ^c^
	GRM + GPS	0.29 ^b,†^	0.30 ^d,†^	0.38 ^a^	0.18 ^a,b^	0.50 ^b^	0.11 ^b^
Marbling Score	GRM	0.40 ^a,†^	-	0.37 ^a^	-	0.35 ^a^	-
	FARM	-	0.12 ^a,†^	-	0.09 ^a^	-	0.07 ^a^
	GPS	-	0.47 ^b,†^	-	0.40 ^b^	-	0.34 ^b^
	GRM + FARM	0.36 ^a,†^	0.10 ^c,†^	0.37 ^a^	0.13 ^c^	0.38 ^b^	0.16 ^c^
	GRM + GPS	0.22 ^b,†^	0.44 ^b,†^	0.37 ^a^	0.39 ^b^	0.63 ^c^	0.35 ^b^

^a^, ^b^, ^c^, ^d^: Different letters indicate statistically different values using Tukey’s multiple comparison test at a significance level of 0.05, comparing the results in each column within each trait. ^†^ Variance component statistically different from zero.

**Table 5 animals-11-02050-t005:** Comparisons of the predicted genomic breeding values (PGBV) between the model that modelled the PGBV alone (GRM) and the models that accounted for the herd effects, either as independent (GRM + FARM), or as correlated (GRM + GPS), for all four traits and all the five models evaluated. The comparisons were performed based on the Pearson ($cor_{P})$ and Spearman ($cor_{S}$) correlations between the PGBVs, and on the mean squared differences between the PGBVs ($MSD_{model\;1,model\;2}=1/n_{test}\overset{n_{test}}{\underset{i=1}{\sum }}{\left({\tilde{g}}_{i,model\;1}-{\tilde{g}}_{i,model\;2}\right)}^{2}$). Values in this table are the mean and standard deviations (in parenthesis) of the comparisons over 100 cross-validation replicates of each model.

Trait	Models Compared		*cor_p_*	$cor_{S}$	MSD
	Model 1	Model 2			
Backfat Thickness	GRM	GRM + FARM	0.98 (0.002)	0.98 (0.002)	0.17 (0.018)
	GRM	GRM + GPS	0.99 (0.001)	0.99 (0.001)	0.07 (0.007)
Eye Muscle Area	GRM	GRM + FARM	0.97 (0.003)	0.97 (0.003)	1.54 (0.167)
	GRM	GRM + GPS	0.99 (0.001)	0.99 (0.001)	0.54 (0.065)
Carcass Weight	GRM	GRM + FARM	0.99 (0.001)	0.98 (0.002)	14.06 (1.438)
	GRM	GRM + GPS	1.00 (0.000)	1.00 (0.001)	3.70 (0.479)
Marbling Score	GRM	GRM + FARM	0.98 (0.002)	0.98 (0.003)	0.02 (0.002)
	GRM	GRM + GPS	0.99 (0.001)	0.99 (0.001)	0.01 (0.001)

## Data Availability

The data that support the findings of this study were available from the Rural Development Administration, Republic of Korea. Restrictions apply to the availability of these data, which were used under license for this study.

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
