# Peer review of "GPS Coordinates for Modelling Correlated Herd Effects in Genomic Prediction Models Applied to Hanwoo Beef Cattle"

_animals, 2021, doi:10.3390/ani11072050_

Round 1
Reviewer 1 Report
Dear authors
This is an interesting paper and makes a valuable contribution to the current body of knowledge. However, as mentioned in the paper the environment in South Korea is very uniform, so to detect environmental effects using only one type of environment is not very accurate. Also please note that environment does not only include the climate, but also the management strategies implemented on the different farms which can also influence the phenotype. I would suggest resubmission after including more different environments if possible. Otherwise you need to describe the environments in Table 1 where you are presenting a summary of the number of farms sampled for the study.
I have made some comments on the article that you need to address as well.

Author Response
AUTHORS: We greatly appreciate the work of the editor and reviewers. The comments and suggestions from the reviewers are valuable to improve our manuscript. We have addressed all the comments from both reviewers, and present here our response to each comment, indicating where the changes/editions are in the revised manuscript, when applicable. We bring here the attention that the line numbering has changed substantially from the original version, due to its adaptation to the journal’s template. The line numbers referred to in our comments correspond to the revised manuscript that we are submitting with the suggested corrections. We also have changed Figure 1, which had the farms located in the map of S Korea, to a figure with only the dispersion of the farms, excluding the map. This change was made due to a request from the data provider.
Reviewer 1
Dear authors
This is an interesting paper and makes a valuable contribution to the current body of knowledge. However, as mentioned in the paper the environment in South Korea is very uniform, so to detect environmental effects using only one type of environment is not very accurate. Also please note that environment does not only include the climate, but also the management strategies implemented on the different farms which can also influence the phenotype.
AUTHORS: Thank you for this comment; indeed, management strategies are important and although we had them in mind, it was missing in the manuscript. We added management strategies as considered part of the environment in our introduction (L60).
I would suggest resubmission after including more different environments if possible. Otherwise you need to describe the environments in Table 1 where you are presenting a summary of the number of farms sampled for the study.
AUTHORS: Although we agree with the reviewer that having a wider range of environment would be of great value to our work, the Hanwoo data available to us with phenotype; genotypes and GPS location is restricted to South Korean producers. Therefore, no different environments can be included for the analyses, other than the S Korean available to us. In the edition of Figure 1, we have generated 3 panels with the dispersion of farms, each panel including a short description of the environments in which the farms are located (altitude, average temperature, and average precipitation). Unfortunately, it was available to us only historical means on temperature and precipitation. In our preliminary study, these historical means showed no statistical significance when included in the model to account for different environments; we believe that was because these historical means do not differentiate the specific environment associated to the records evaluated on their specific time period (2014-2017).
I have made some comments on the article that you need to address as well.
L15: what is meant by them?
AUTHORS: ‘them’ was meant to be ‘environmental effects’. We clarified this in the manuscript (L13).
L20-23: Be clearer. L21: ? Not sure what impacts you are referring to here.
AUTHORS: We have revised this sentence to make it clearer (L18-19).
L23-25: Give the parameters for accuracy in values with standard errors.
AUTHORS: We added the increases in reliability with the SE’s (L21-22).
Keywords: herd effects is already mentioned in the title
AUTHORS: Thank you for noticing, we removed ‘herd effects’ from the key words. Since the title was edited, we revised and edited all the key words.
Introduction: the introduction is not adequately written, more literature is needed to give a complete background for the study.
AUTHORS: We have revised the introduction to include the references needed and included extra editions to strengthen the background.
L33: Which modern approached are you referring too?
AUTHORS: We refer to genetic evaluations, this is now clear in the manuscript (L29).
L37-40: give a reference for this statement.
AUTHORS: We added a reference to the statement (L35).
L57-59: why? elaborate on this.
AUTHORS: We have extended the introduction on this part (L50-57).
L84-86: put line 84-86 in the materials and methods section.
AUTHORS: Thank you for the suggestion, this sentenced was moved to M&M (L86-87).
L88 -90: include the ethical clearance number for the study.
AUTHORS: The approval number of the main project which comprises the data used was included (L83).
L170-171: which specific software did you use here?
AUTHORS: This information was at the end of section 2.3; we have moved it to the beginning of the section so that it is immediate for the reader (L155-156).
L219-220: Normally you would use the maximum likelihood value to determine the most accurate models and not a tukey's test.
AUTHORS: Indeed, likelihood-based methods – such as a likelihood ratio test (LRT), or the Akaike information criterion (AIC) – are traditionally applied to compare model fitness. However, in our study we did not aim to verify which model obtained a better fit of the data, but to verify which model obtained a better prediction accuracy for the predicted breeding values. The assessment of the models’ predictability is done by cross-validation, i.e. the test group does not figure in the training step of the model. Thus, no likelihood values can be derived on the test group. In this context, we performed the comparison of means (over the 100 replicates) of the prediction accuracies/reliabilities. Likewise, we performed the comparison of means (over the 100 replicates) of the heritabilities and environmentabilities to evaluate significant differences in the parameters. Since we wished to perform the comparison over more than two models applied, Tuckey’s test was used to ensure correct statistical significance on the multiple comparison tests.
L226-229: rephrase this sentence, rather start with the heritability estimates for EMA, CWT etc...
AUTHORS: We have rephrased the sentence to be as suggested by the reviewer (L204-206).
L327: accurate instead of correct.
AUTHORS: Thank you for the suggestion, we have replaced ‘correct’ by ‘accurate’ (L264).
L334-338: reference?
AUTHORS: We added references to this statement (L269-273).
Table 1: Give an indication of the different environments on the different farm locations.
AUTHORS: We have included the information available to us about the different environments on Figure 1.
Table 2: add the number of records used under each section, for bull, steer and all throughout the table (N=?).
AUTHORS: We have added the number of animals detailed by age and sex in Table 2.
Table 4: justify text of footnote.
AUTHORS: This has been corrected to be right justified.
Table 5: describe CorP and CorS better in the caption.
AUTHORS: We edited the caption to clarify these.

Reviewer 2 Report
Dear Authors,
I have reviewed the manuscript entitled 'GPS coordinates for modelling correlated herd effects in genomic prediction models' by Cuyabano et al.
The manuscript presents a study on the advantage of including GPS coordinates to impose a covariance among herds in Hanoi Korean cattle. The study is definitely of interest and is based on a solid theoretical background. The cross-validation is well constructed and the Authors also checked for collinearity among the random effects structures, which is something that more people should do. The manuscript is well written. The findings are readily applicable, easy to implement and could actually lead to some changes in the genetic evaluations.
In general, I would not use the term 'relationship matrix' when it comes to herds. I'd just use the term 'covariance matrix'. This is because the term 'relationship', in animal breeding, has the specific meaning of sharing alleles.
I suggest referring to the subject population, or at least the species, in the title.
PBGV is not defined. Would that stand for 'predicted genomic breeding values'?
I want to suggest the Authors to consider this paper (https://doi.org/10.3168/jds.2016-11543) in their discussion. A kernel built on GPS coordinates was also used, among the others.
Some in-text comments:
Line 58: 'as random effect in models'.
Line 129: I'm usually skeptical of the use of age as an adjustment in the model, because age of slaughter can be used as a preferential treatment. However, as the Author mention a preliminary study, they have probably consider this point already.
Line 203: '2018' should probably be between parentheses.
Author Response
Reviewer 2
Dear Authors,
I have reviewed the manuscript entitled 'GPS coordinates for modelling correlated herd effects in genomic prediction models' by Cuyabano et al.
The manuscript presents a study on the advantage of including GPS coordinates to impose a covariance among herds in Hanoi Korean cattle. The study is definitely of interest and is based on a solid theoretical background. The cross-validation is well constructed and the Authors also checked for collinearity among the random effects structures, which is something that more people should do. The manuscript is well written. The findings are readily applicable, easy to implement and could actually lead to some changes in the genetic evaluations.
In general, I would not use the term 'relationship matrix' when it comes to herds. I'd just use the term 'covariance matrix'. This is because the term 'relationship', in animal breeding, has the specific meaning of sharing alleles.
AUTHORS: We have edited ‘herd relationship matrix’ to ‘herd covariance matrix’ throughout the entire manuscript. Conceptually, a covariance represents a relationship, and therefore the term ‘herd relationship matrix’ is not necessarily wrong. However, we agree that as pointed out by the reviewer, in the context of animal breeding, the term ‘relationship’ tends to be interpreted in a concept of allele sharing, and ‘covariance relationship matrix’ is more adequate to use in the manuscript.
I suggest referring to the subject population, or at least the species, in the title.
AUTHORS: Thank you for this suggestion. We agree that given the special issue to which we submitted this manuscript, adding the population/species to the title is important. Therefore, the title has been edited to this reference.
PBGV is not defined. Would that stand for 'predicted genomic breeding values'?
AUTHORS: Thank you for noticing. This is now fixed (L162) at the first mention of PGBV.
I want to suggest the Authors to consider this paper (https://doi.org/10.3168/jds.2016-11543) in their discussion. A kernel built on GPS coordinates was also used, among the others.
AUTHORS: Thank you for the reference suggestion. We have read the paper, and although the context of their use of location data is different than ours, it is a relevant citation in the sense that it uses a set of information to correlate environmental effects included in the GP model. We included its citation in the introduction (L62-64).
Some in-text comments:
Line 58: 'as random effect in models'.
AUTHORS: Thank you for noticing the missing word, we have added it (L50).
Line 129: I'm usually skeptical of the use of age as an adjustment in the model, because age of slaughter can be used as a preferential treatment. However, as the Author mention a preliminary study, they have probably consider this point already.
AUTHORS: We agree that age can be deceiving in beef cattle studies. However, our data was provided by finishing farms, .where no preferential treatment related to the age is expected.
Line 203: '2018' should probably be between parentheses.
AUTHORS: Thank you for noticing. We have corrected it (L181).